# Ligno-Cellulosic Fibre Sized with Nucleating Agents Promoting Transcrystallinity in Isotactic Polypropylene Composites

**DOI:** 10.3390/ma13051259

**Published:** 2020-03-10

**Authors:** Armin Thumm, Regis Risani, Alan Dickson, Mathias Sorieul

**Affiliations:** Scion, Forest Research Institute Ltd., 49 Sala street, 3020 Rotorua, New Zealand; Armin.thumm@scionresearch.com (A.T.); Regis.risani@scionresearch.com (R.R.); Alan.dickson@scionresearch.com (A.D.)

**Keywords:** nucleating agent, isotactic polypropylene, transcrystallinity, natural fibres, Tencel™

## Abstract

The mechanical performance of composites made from isotactic polypropylene reinforced with natural fibres depends on the interface between fibre and matrix, as well as matrix crystallinity. Sizing the fibre surface with nucleating agents to promote transcrystallinity is a potential route to improve the mechanical properties. The sizing of thermo-mechanical pulp and regenerated cellulose (Tencel™) fibres with α- and β-nucleating agents, to improve tensile strength and impact strength respectively, was assessed in this study. Polarised microscopy, electron microscopy and differential scanning calorimetry (DSC) showed that transcrystallinity was achieved and that the bulk crystallinity of the matrix was affected during processing (compounding and injection moulding). However, despite substantial changes in crystal structure in the final composite, the sizing method used did not lead to significant changes regarding the overall composite mechanical performance.

## 1. Introduction

Isotactic polypropylene (iPP) is an important engineering thermoplastic polyolefin used in many different commercial applications such as packaging and automotive parts [1]. The main advantages of iPP are its easy processing and low manufacturing cost [2]. Its mechanical behaviour, thermal properties, and chemical resistance depend on its semi-crystalline structure and fraction (typically 50%–70%) [3]. Isotactic polypropylene is a non-polar, polymorphic polymer. The three basic crystal forms of iPP are: the monoclinic alpha (α) form, the trigonal beta (β) form and the orthorhombic gamma (γ) form [4,5]. The α-form crystals are obtained under common processing conditions, between 60 °C and 188 °C, with a maximum crystallisation rate around 80 °C [6,7]. In composites, the presence of α-crystals improves thermodynamic stability and mechanical performance but also reduces the impact strength, especially at low temperature. The β-form crystals are metastable thermodynamically, and have several advantages over the α-form, such as improved elongation at break and improved impact strength [8,9]. The β-crystals fan-shaped morphology has the ability to dissipate the impact energy, therefore improving toughness [10]. The β-form can be obtained using a temperature gradient method [11], shear flow-induced crystallization [12,13] or specific nucleating agents (NA) [14]. A temperature of crystallisation around 130 °C is best to promote nucleation and growth of β-iPP crystals [15,16]. 

It is a common strategy to improve the mechanical properties of polyolefin-based materials via the introduction of reinforcing agents (e.g., carbon fibres, glass fibres, clays, lignocellulosic materials). The strength of composite materials is influenced by: matrix properties, intrinsic properties of the reinforcement, dispersion of the reinforcement in the matrix, degree of orientation, quality of the interface and volume fraction of reinforcement [17,18]. In the resulting composite, the stress concentration develops at the interface between the matrix and reinforcement agent and is mainly influenced by the following: the volume fraction; the thermal expansion coefficient difference between each material; the interphase; and the crystallisation of the matrix. To achieve high mechanical performance, good interfacial adhesion allowing stress transfer between the matrix and reinforcement agent is crucial [19,20]. Several methods have been developed to improve interfacial adhesion between highly polar fibres and nonpolar polymers. The most common strategies used are sizing, compatibilisation, polymer grafting and interfacial crystallisation [10]. Transcrystallinity (TC) or transcrystalline layers describes a crystalline structure with limited thickness located at the interphase region, it originates from a high density of heterogeneous nuclei with a crystal growth orientation mainly perpendicular to the fibre surface until the growing front impinges with spherulites nucleated in the bulk [21,22]. The development of a TC structure is a promising route for improving the load transfer between the semi-crystalline matrix and the fibre [23].

TC enhances fibre–matrix adhesion, reduces stress concentration, creates a mechanical interlock and a protective layer around the fibres leading to an efficient stress transfer from the matrix across the interphase [24]. Good TC is obtained when the crystallisation parameters are more favourable to the TC formation compared to the crystallisation of spherulites in the bulk [25]. In composites, the development of the TC layer is influenced by factors such as matrix type, thermal history, temperature of polymer crystallisation, chain mobility, rate of cooling, occurrence of shearing forces during crystallisation and thermal expansion coefficients of individual components [24,26]. The surface topology, composition and sizing of the fibre itself are major parameters influencing the TC formation [27]. To achieve good TC, high nucleation site density along the fibre surface is crucial [28]. The proximity of nucleation sites restricts the crystal growth to the lateral direction, leading to the development of a columnar layer around fibres [22,29].

Lignocellulosic fibres have many beneficial features such as biodegradability, renewability, availability, low cost, and ease of preparation [30]. Polymer composites incorporating lignocellulosic fibres offer several advantages including low density, good mechanical properties and high damping capabilities [31,32]. The lignocellulosic fibres chosen for this study were High Temperature Thermo-Mechanical Pulp fibres (HT TMP) and Tencel™. The HT TMP fibres are naturally coated with a thin, relatively non-polar, lignin layer [33,34,35]. Softwood HT TMP fibres are of industrial relevance due to their low cost of production and low weight advantage when compounded into plastic composites [36,37]. Tencel™ is a regenerated cellulose fibre which exhibits high tensile properties (strength of 1.4 GPa and modulus 36 GPa) [38] and was included in this study as a reference fibre, as its high ductility improves the impact characteristics of brittle polymer matrices [39].

This study investigates the potential of α- or β-nucleating agent (NA) as a sizing additive for improving the mechanical properties of an iPP-natural fibre composite. The hypothesis is that the NA will generate a high nucleation density on the fibre surface leading to an α or β specific TC, improving both tensile properties and/or impact resistance. The addition of the NA usually occurs at the compounding stage of composite manufacture. In this study, we developed an original approach by directly adding the NA onto the fibres that could be adopted to tailor the lignocellulosic fibre composite characteristics eliminating the need for the addition of NA at the compounding stage. Our novel approach was to coat the fibres with the NA prior to compounding with the iPP to create a TC structure with improved matrix-fibre stress transfer.

## 2. Materials and Methods 

### 2.1. Materials

#### 2.1.1. Fibres 

Wood chips (*Pinus radiata*), were thermomechanically pulped at 180 °C at Scion’s fibre and pulp processing pilot plant (Rotorua, New Zealand). The resulting HT TMP fibres were dried in a tube drier to approximately 12% moisture content. An earlier study has found that chips processed in this way lead to fibres with a cellulose content of 39%–42%, lignin content of 28%–31% and 22%–25% hemicelluloses [40]. Prior to processing the fibres were approximately 1.25 mm long and 0.03 mm wide. The length and aspect ratio were significantly reduced during compounding [41]. Tencel™ fibres were purchased from Lenzing Fibers GmbH (Heiligenkreuz, Austria). 

#### 2.1.2. Nucleating Agents

The α-NA was Hyperform^®^ HPN-68L (Milliken Chemical, Blacksburg, SC, USA) and the β-NA was NU-100 (NJStar), (New Japan Chemical Co. Ltd., Osaka, Japan). Hyperform^®^ HPN-68L is a powdered α-NA comprising a dicarboxylate sodium-based compound known as HPN-68L. It is used to reduce cycle time and improve the tensile and flexural mechanical properties of the iPP [42]. Nu-100 (NJSTAR NU-100) is a β-NA. Its active ingredient, N,N′-dicyclohexyl-2,6-naphthalenedicarboxamide (DCNDCA), has been found to be an effective β-NA [15]. However, DCNDCA is dual selective and can induce both α-phase and β-phase, depending on the thermal condition applied [15].

#### 2.1.3. Polypropylene and Compatibiliser 

Isotactic homo-polypropylene SJ-170, Hopelen was sourced from Lotte Chemical Co., Seosan, South Korea. The compatibiliser was 3% w/w maleic anhydride grafted polypropylene (MAPP) Epolene G3015 (Eastman Chemical Co., Kingsport, TN, USA). A 3% MAPP loading was used according to the manufacturer recommendation for 30% wt lignocellulosic fibre content.

### 2.2. Methods

#### 2.2.1. Sizing of the Fibre and Fibre Pellet Production

Fibres were blended with an adhesive thermoplastic acrylated emulsion [36] and NA in a dry blender consisting of a 12 m steel loop (Ø: 0.15 m) with a fan that creates a turbulent flow. The adhesive was administered onto the fibre in the loop via an inserted spray gun. The NA powder was then slowly added into the loop and the materials were circulated in the loop for a further 2 min. The adhesive was added at a 4% wt loading with respect to oven-dried fibre. The NA was added at a 1.7% loading with respect to fibre (0.5% wt loading with respect to the finished composite). Coated fibres were hot-pressed into 3 mm thin sheets and subsequently cut into small dice (4 × 4 mm) with a pneumatic chopper. The process is described in detail in Warnes et al., [36]. 

#### 2.2.2. Nitrogen Analysis

Between 0.25 and 0.5 g of fibre dice were heated in a stream of high purity oxygen in a Leco furnace (Laboratory Equipment Corporation, St Joseph, MI, USA) to produce CO_2_, N_2_ and NO_x_. A subsample of the combustion gases was passed through a heated copper catalyst that further reduced the NO_x_ to N_2_, which was then measured by thermal conductivity. This results in the percentage of total nitrogen. This percentage was converted to NA content of the final sample by subtracting the N content of a NA free reference from the sample containing NA and then dividing the resulting difference by the nitrogen content of the pure NA. This method could only be applied to the β-NA as the α-NA does not contain nitrogen. 

#### 2.2.3. Compounding

The HT TMP fibre dice and the Tencel™ fibre dice were compounded at 30% wt loading into iPP with 3% wt maleic anhydride PP (MAPP) as a coupling agent. The LabTech™ extruder (LTE26-40, LabTech Engineering Co. Ltd., Samut Prakan, Thailand) had a 26 mm screw diameter, with co-rotating twin screws, with a 40 L/D (length/diameter) ratio. The PP and MAPP were dry blended and fed into the main feed throat using a Weighbatch™ DS20 (Weighbatch, Hamilton, New Zealand) gravimetric feeder. The fibre dice, which were oven dried overnight at 105 °C, were side-fed into the extruder using a K-Tron twin-screw gravimetric feeder (Coperion K-Tron, Sewell, NJ, USA). For the “no fibre” reference, the NAs were dry-blended with the iPP at the same time as the MAPP. Two atmospheric venting ports and one crammer vacuum (0.7 bar) degassing port were used to remove the entrapped air in the melt along with volatile organic compounds (VOCs). The melt was extruded through a 2-strand die and pulled into a water bath before being granulated into 3 mm long pellets. Each formulation was compounded in a single extrusion run at 200 rpm screw speed and 8 kg h^−1^ total extrusion throughput to ensure proper fibre mixing with a gentle screw design and reverse barrel temperature profile (220 °C to 190 °C).

#### 2.2.4. Injection Moulding

Prior to injection moulding, the compounded pellets were oven dried at 105 °C, to obtain a pellet with residual moisture content below 0.3% wt. The compounded pellets were injection moulded with a BOY 35 machine (BOY Spritzgiessautomaten, Neustadt-Fernthal, Neustadt-Fernthal, Germany) into ISO multipurpose injection moulded test specimens (dogbone) type A (ISO 3167). The barrel temperature of the injection moulder was 190 °C and the mould was kept at 30 °C. The injection speed was 100 mm s^−1^. The mould was filled with a screw speed of 100 rpm with 15 bar back pressure. Cooling time was 20 s. The injection moulding parameters were the same for all composites. Injection moulded parts were collected once their weight was constant. Thirty test specimens were collected for each treatment with the first ten disregarded for analysis. 

#### 2.2.5. Polarised Light Optical Microscopy (PLOM)

Cross-polarized optical microscopy was performed with a Leica DMRB microscope (Leica Mikroskopie & Systeme GmbH, Wetzlar, Germany), using a Leica EC3 camera (Leica Microsystems Ltd., Singapore) and a 10× magnification lens (Leica PL Fluotar). The temperature was controlled during the experiments with a programmable Mettler-Toledo hot stage HS82 (Mettler-Toledo GmbH, Greifensee, Switzerland).

For each experiment, a piece of iPP thin film was placed on a glass microscopy slide. Individual fibres are then positioned onto the film. The slide was placed into the temperature-controlled stage [43]. The stage temperature was raised to 200 °C and once the iPP was molten, the sample was covered with a cover slip. After a 5-min holding period at 200 °C, to erase the thermal history of the sample, the temperature was decreased with a 10 °C/min ramp and stabilized at 133 °C to monitor the isothermal crystallisation (Scheme 1). The samples were imaged after 5 min of isothermal crystallisation at 133 °C. The temperature of 133 °C was chosen as it is the optimal temperature to promote β over α crystallisation [16,25].

#### 2.2.6. Etching

Injection moulded samples were etched based on the method of Olley [44]. This method removes any amorphous material so that the remaining crystalline structure could be observed by electron microscopy. A 35 mm section was cut from the middle part of a tensile specimen. The section was gently stirred for 6 h in a solution of 1.3% wt potassium permanganate (KMnO_4_), 32.9% wt concentrated sulfuric acid (H_2_SO_4_) and 65.8% wt concentrated phosphoric acid (H_3_PO_4_) at room temperature. It was then cleaned with hydrogen peroxide for 5 min, rinsed with water, and dried overnight in an oven at 105 °C.

#### 2.2.7. Scanning Electron Microscopy (SEM)

All SEM samples were coated with chromium using an Emitech K575X sputter coater (Quorum Technologies Ltd., Kent, UK) and imaged using a JEOL JSM 6700F (JEOL Ltd., Tokyo, Japan) at 3 kV accelerating voltage. For publication purposes the images were contrast enhanced using the enhanced local contrast (CLAHE) plugin in the ‘Fiji’ distribution of ImageJ (V1.5h) [45].

#### 2.2.8. Crystal Size Measurement

The SEM images generated from the etched samples were analysed using V++ software (Digital Optics, Version 5.0, Wellington, New Zealand) to determine the diameter of the crystals. Approximately 50 crystals were measured for each treatment. 

#### 2.2.9. Differential Scanning Calorimetry (DSC)

For DSC analysis, all measurements were performed in triplicate. A transverse section (0.35 mm) was cut from the middle part of a tensile specimen. To avoid measuring the crystallinity of the skin of the test specimens, only the core of the transverse section (~5 mg) was used. All the calorimetric experiments were performed with a Discovery (TA instruments, USA) differential scanning calorimeter under nitrogen atmosphere (50 mL min^−1^). The temperature scale was calibrated using indium, lead and tin as standards to ensure reliability of the data obtained. Melting temperatures, enthalpies, crystallisation and fusion peaks were determined by TRIOS software (TA Instruments, USA). The degree of crystallinity (Ӽc) was estimated using Equation (1) where ΔHm is the measured melting enthalpy of the polymeric part of the sample, Wf the fibre weight fraction in the composite, and ΔH 100% the equilibrium melting enthalpy of 100% crystalline PP assumed to be 207 J g^−1^ [46].
(1)$$Ӽc=100\;\times \;\frac{\Delta \mathrm{Hm}}{\Delta H\;100\%\;\left(1-\mathrm{Wf}\right)}$$

The thermal gradient of the DSC measurements is described in Scheme 2.

The enthalpy variations of the polymer during the initial heating ramp (Scheme 2), gives an indication of the thermal history related to the injection moulding conditions. A maximum temperature of 190 °C was used to limit the natural fibre degradation but, as a consequence, the thermal history might not have been totally erased. This step is followed by a slow cooling ramp at a rate that favours the formation of a high β-fraction [15]. The melting cycle observed during the second heating ramp illustrates the crystal formation process which occurred during a controlled slow cooling with limited thermal history.

#### 2.2.10. Mechanical Testing

##### Tensile and Modulus Tests

Tensile properties were measured on an Instron 5566 (Instron, Norwood, MA, USA) universal testing machine fitted with a 10 kN load cell and an external extensometer, according to ISO 527. The crosshead speed was 5 mm min^−1^. The gauge length was 115 mm and the extensometer length were set to 25 mm. Ten specimens were tested to failure to obtain the average Young’s modulus and maximum tensile stress.

##### Impact Test

Samples for Izod notched impact strength were prepared and tested in accordance with ASTM D 256. Samples were of dimensions 12.6 mm × 63.5 mm x thickness and a 45° notch was machined into each sample using a Ceast Notchvis (Ceast, Italy). Seven Izod samples for each variable were tested at a velocity of 3.46 m s^−1^, a 150° angle and a 0.5 J hammer using a Ceast Resil impact testing machine (Ceast, Torino, Italy). Pull-out surfaces were compared between different formulations with SEM pictures taken from the central part of the transverse fracture area.

#### 2.2.11. Water Uptake

ISO test specimens (dogbone) were submerged in water at 20 °C for 70 days. During this time water uptake was monitored by briefly removing the samples from the water bath, wiping the surface dry with a tissue and weighing them. This approach is based on ASTM D1037. Samples were measured in triplicate.

## 3. Results

### 3.1. Crystal Structure

The PLOM analysis showed no obvious transcrystallinity around the fibres without NA (Figure 1A,D). Although transcrystallisation is generally expected in cellulose fibres [47,48,49,50], it is also known to be impeded by the presence of lignin and hemicelluloses that decrease the frequency of nucleation sites and degree of crystalinity [51,52]. The absence of transcrystallinity with the untreated Tencel™ fibres is likely due to the cellulose II configuration of these artificially produced fibres which reduces nucleation activity [53,54,55,56].

Treatment with NA resulted in crystal growth perpendicular to the two types of fibres (Figure 1B,C and Figure 1E,F). The fibres treated with the β-NA (Figure 1C,F) showed better definition of the crystalline structure than those treated with the α-NA (Figure 1B,E). 

SEM of the composites after compounding and injection moulding confirmed that the iPP matrix containing fibres treated with the β-NA had iPP crystals that were fan-shaped, typical of β-crystals, whereas the samples without β-NA only showed α-crystal morphology (Figure 2). The etching process not only removes the amorphous phase of PP but also the fibres from the composite. The images show that the crystals were not specifically present around the space previously occupied by the fibre but seemed to be distributed throughout the matrix.

Compared to the controlled PLOM analysis, the compounded and injection moulded composites investigated by SEM were subjected to shear stresses and faster cooling. Large α-crystals were evident in the pure iPP (Figure 2a). The introduction of HT-TMP (Figure 2d) and Tencel™ (Figure 2g) fibres did not change the crystal type and morphology but their size was decreased by half (Figure 2j). This was the result of steric hindrance as the presence of the fibres increased the number of nucleation sites thereby increasing the crystal density and thus restricting the growth of the α-crystals. The α-NA resulted in a 30-fold size reduction of the α-crystals. This is likely due to the high number of nucleation sites leading to a growth competition. The addition of both types of fibres (Figure 2e,h) did not lead to further reduction of the crystal size (Figure 2j). The fan shaped β-crystals induced by the β-NA (Figure 2c) had an average diameter around 3.3 µm (Figure 2j). The addition of HT TMP and Tencel™ fibres (Figure 2c,f) also did not reduce the size of β-crystals further (Figure 2j). The hollows in the centre of β-spherulites are attributable to the etched DCNDCA particles [57].

### 3.2. Differential Scanning Calorimetry (DSC)

Total crystallinity for all samples ranged between 36% and 45% (Table 1). The addition of fibre to the iPP had little effect on the overall crystallinity of the composites. The addition of NA also had little effect on the overall crystallinity of the iPP, however, the type of NA had a significant effect of the type of crystal structure present. As expected, no β-crystals were observed in the composites without the addition of β-NA. 

The addition of β-NA, in the absence of fibre, led to an approximately 13% reduction of α-crystal enthalpy. The decrease in the proportion of α-crystals was compensated by the appearance of β-crystals. Therefore, the overall crystallinity was maintained. The same trend was observed in the presence of both types of fibre. Removal of the thermal history led to a larger decrease in α-crystals, down to ~10.5% and the highest level of β-crystals observed 34% and 30.4% for HT TMP and Tencel™, respectively. The presence of fibres coupled with a slow cooling ramp and without shear stress was the most favourable condition observed to generate a high β-crystal proportion. Our results are in accordance with Kang et al., who found that iPP in the presence of the dual selective β-NA (TMB-5) coupled and a slow cooling rate favoured the formation of high β-crystal fraction [15]. However, those results contradict Dong et al., [57] who found that fast cooling is favourable for creating high β-content in iPP/TMB-5 system. In our case, the slow cooling was favourable to the growth of β-crystals, possibly because the β-crystals formed at a higher temperature compared to α-crystals and, therefore, β-crystals had time to grow without competition from the α-crystals.

Evaluating how a combination of a slow cooling ramp and high shear stress (injection moulding) affects β-crystals formation was not technically possible. Indeed, fast cooling of the polymer is central to the injection moulding process to reduce cycle time without compromising on part quality [58].

The potential factors explaining why the composites containing HT TMP fibres had a higher β-crystals content compared to one with Tencel™ fibres are probably related to the fibre surface. Tencel™ fibres have a smooth surface [59] and HT TMP fibres are rough with a coating of lignin. A rough surface increases the interfacial shear strength [60] which is a factor favouring β-interfacial structures [61]. Moreover, lignin particles can increase β-crystallinity [62]. 

The addition of fibre into iPP without NA did not have a major influence on the melting temperature of the polymer, neither did the addition of the α-NA (Table 1). The melting temperature was around 163 °C for the α-crystals for all composites. The addition of the β-NA added a second melting peak at 152 °C (after removal of thermal history). For the samples with β-NA, the cooling process slightly shifts the β-crystals melting peaks.

Compared to IM, after a 10 °C min^−1^ cooling ramp, the melting peak temperature of the β-crystals increases by 2.6 °C, 5.1 °C and 4.9 °C for the iPP with β-NA without fibres, with HT TMP and with Tencel™ respectively. This indicates that the β-crystals which were grown at high temperature during slow cooling are more stable and therefore melt at higher temperature [63].

Samples were cut from the centre of the transverse surface of a tensile test specimen (dogbone). Proportions are extracted from integration of the melting peaks obtained from the DSC experiment. (n = 3, ± = standard deviation)

For industrial processing, the crystallisation temperature is an important parameter as a higher crystallisation temperature allows faster processing [64]. The presence of fibre in the iPP increases nucleation sites, and thus the crystallisation temperature by 6 and 8 °C for the HT TMP and the Tencel™, respectively (Figure 3).

Addition of α-NA does not affect the proportion of α-crystals in the iPP, but its crystallisation temperature is 20 °C higher. This is lowered with the addition of fibre. In presence of α-NA, the crystallisation temperature for HT TMP and Tencel™ composites increased by 12 and 8 °C, respectively. The addition of β-NA leads to a 15 °C rise in crystallisation temperature for the pure iPP. Again, this was lowered with the addition of fibre. The presence of β-NA lead to 8 and 5 °C increase in crystallisation temperature for the HT TMP and Tencel™ iPP composites, respectively. Overall, the pattern of crystallisation temperature increase remains the same for composites filled with either fibre, with a minor antagonist effect when the NA and fibres are present together.

### 3.3. Mechanical Properties

For commercial application, obtaining a composite with good mechanical properties especially yield strength and impact strength is extremely important [58]. 

#### 3.3.1. Young’s Modulus and Stress

The addition of fibre to iPP led to at least a doubling of the Young’s Modulus (Figure 4a). The effect of the NA alone is minimal in comparison, with an increase of 29% and 17% for α-NA and β-NA added to iPP. Additionally, the addition of both NAs in presence of fibres has no significant effect on Young’s Modulus of the composites with the exception of α-NA with Tencel™ (+18%).

The situation is relatively similar for tensile maximum stress properties (Figure 4b). The addition of β-NA has no effect on iPP, while the α-NA leads to a moderate increase (+13%). The addition of HT TMP and Tencel™ generates an increase in tensile maximum stress of 46% and 75%, respectively. In all cases, the addition of α-NA lead to a minor increase, while the β-NA lead to an expected small decrease in tensile maximum stress. 

#### 3.3.2. Impact Strength

Reinforcing iPP with HT TMP fibre gave an improvement in impact strength of the composite (Figure 4c). Adding an impact strength modifier, such as a β-NA, to the fibre was considered an opportunity to further enhance impact performance. Compared to pure iPP, the sole addition of HT TMP fibre led to a 46% increase in impact strength. When added in conjunction with HT TMP fibre, the α-NA addition reduces the impact strength of the fibre composite compared to the addition of fibre alone. β-NA has no significant influence. The major gain is due to the presence of HT TMP. A major improvement in impact strength comes from the addition of Tencel™ fibres with an increase of 181%. Addition of α-NA led to no significant change, while the addition of the β-NA led to moderate further improvement (+17%).

In summary, the main improvement for mechanical properties comes from the presence of the fibres themselves. In some cases, the NAs are providing a further minor improvement but in other cases, they can also have detrimental effects. 

### 3.4. Pull Out

SEM micrographs of transverse fracture surfaces from tensile testing samples show only minor pull-out of fibres for HT TMP filled composites (Figure 5). The few visible fibres appear to be delaminated or covered in matrix. The influence of lignin is complex, although lignin removal was found to improve composite properties [65], the presence of lignin can also have the effect of improving properties due to its interaction with coupling agents [66,67,68,69]. There are no visually observable differences in pull-out between the different HT TMP treatments. The fracture surfaces of Tencel™ filled-composites show a large number of fibres being pulled-out across all treatments. No matrix material adheres to the pulled-out fibres. It indicates that the interfacial adhesion between Tencel™ and iPP is either lower than that of HT TMP and iPP or it can be explained by their higher shear strength compared to the HT TMP fibre [70,71]. Those observations agree with the higher impact strength observed for the Tencel-based composites, as a composite containing high strength fibres weakly compatibilized with the matrix will exhibit good impact performance. The TC layer observed by PLOM in the sandwich composite is clearly not covalently attached to the Tencel™ fibres after injection moulding. The NA do not seem to have a strong effect on the interface.

### 3.5. Water Uptake

A slow water absorption rate allows for better dimensional stability of the composite when exposed to humidity. The nine formulations were immersed in water for 70 days. The results are expressed in percentage of water absorbed (Figure 6 and Table 2). The water uptake of the pure hydrophobic iPP is relatively low (0.3%). This value is comparable to previous studies [72]. The addition of β-NA leads to a slight but significant decrease in water absorption. 

The hydrophilic character of natural fibres, ascribed to the bonding of water molecules to the free hydroxyl group is responsible for a water uptake increase in the plastic-fibre composites [73]. The addition of HT TMP and Tencel™ to iPP lead to an increase of water absorption to around 2%. This indicates that the two types of fibres have the same type of hydrophilic behaviour. Surface treatment of fibre has often been used to decrease natural fibre composite water uptake [74,75]. Wu et al., [76] compared the water absorption of flax fibres functionalised with MAPP within either α or β crystalline iPP matrix. They did not observe any difference between the two types of crystal matrix. However, when the fibres were not functionalised, the α matrix containing the flax fibres had a slightly higher water absorption than its β crystals counterpart. In our case, the addition of β-NA similarly shows a small but significant reduction in water uptake that is consistent across all treatments. 

Unexpectedly, the addition of α-NA shows a significant increase in water uptake for samples containing fibres. The increase is only 0.5% in the case of the slightly hydrophobic HT TMP fibres, but is more pronounced (3.4%) in the case of the hydrophilic Tencel™ fibres. 

Additionally, DSC analysis (shown earlier) indicated the crystallinity of the α-NA and fibre composites was lower than both the control and β-NA composites which is correlated to this increase in water uptake. This increase in amorphous structure will create pathways for accelerating the water penetration into the composite. The larger the decrease in crystallinity, the larger the water uptake

## 4. Discussion

The main aim of this study was to improve the mechanical properties of HT TMP composites. We developed an original approach by directly adding the NA on the fibres which could be adopted to tailor the lignocellulosic fibre composite characteristics without an additive incorporation step. The sizing of the fibres was successful; however, the hoped-for tensile strength (α-NA) and impact resistance (β-NA) improvements were not observed. Our strategy failed to improve the mechanical properties of the composite. It may be that the interface between lignocellulosic material and polyolefin matrix in presence of MAPP is already sufficiently strong and is not improved by the TC and nucleation of the matrix [77]. The absence of covalent linkage between the NA and the fibre might be one of the reasons for the lack of mechanical properties improvement. Another reason could be that the concentration of β-NA is too high. To allow for losses in the blending process, a relatively high concentration of NA was applied. However, losses were modest and nitrogen analysis revealed the concentration of β-NA in the final composite to be 0.4%. Other studies have reported an optimal concentration of β-NA is 0.1%. This allows a maximal growth of β-spherulite in needle like structure and a concentration as low as 0.2% would lead to β-crystals growth competition leading to incomplete spherulite and a decrease on impact strength [78]. When a composite has a high concentration of β-NA, the ordered structure effect cannot be reached [79]. Another aspect could be that the crystal structure at proximity of natural fibre might impair the MAPP efficiently grafting onto the fibre.

## 5. Conclusions

This study investigated the effect of coating lignocellulosic fibres with NA on the mechanical properties of the resulting iPP-fibre composite. The influence on the TC layer, crystal morphology and proportion, crystallisation temperature, mechanical properties, surface fracture and water uptake were investigated. The addition of NA led to slight increases in composite crystallinity as well as an increase in temperature of crystallisation. The presence of the NA also leads to a significant decrease in crystal size. Sizing the fibres with NA was shown to be an effective method to manipulate the crystallinity of the natural fibre composite. The presence of NA is deeply modifying the nature of matrix crystallinity and its thermal characteristics. However, under the experimental conditions of this study, the mechanical properties of the iPP-fibre composites were mainly influenced by the characteristics of the fibres while the NA had little influence. This suggests that the NAs probably have little influence on interfacial adhesion. In summary:The α- and β-NA as fibre sizings have the expected effect on iPP crystals.The α- and β-NA cause transcrystallinity at the interphase fibre-matrix in a sandwich composite.The α- and β-NA affects the whole matrix crystallinity after compounding.The tensile strength and impact strength of the fibre-iPP composite is not significantly improved by the NA.

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
