# Peer review of "Ligno-Cellulosic Fibre Sized with Nucleating Agents Promoting Transcrystallinity in Isotactic Polypropylene Composites"

_materials, 2020, doi:10.3390/ma13051259_

Round 1

Reviewer 1 Report

The manuscript by Thumm et al. describes polypropylene/natural fibers composites. This study covers an interesting subject and fits well current trends regarding use and new applications of lignocellulosic materials. The article is well-written, materials and methods are nicely described. My opinion is that this article after minor revisions can be published in the Journal, however some questions should be addressed:

Section 2.2.1: What kind of adhesive was used?

Section 2.2.6: Please use subscripts in chemical formulas.

Section 3: Please provide information regarding induction time and growth rate of spherulites. Did Authors perform XRD studies of composites and fillers? It would give some insight into supermolecular structure of materials. If Tencel fibers are regenerated, does it mean they contain cellulose II? The Authors are also encouraged to compare their results with more literature data.

Line 283: Are Authors sure that there was no β phase in samples after injection molding?

Line 296-298: Authors claim that slow cooling was favourable in terms of β phase formation. Did Authors try to carry out an isothermal crystallization in higher temperature?

Line 308: I agree that lignin can influence crystallization of iPP. However, it is not so obvious – please compare with e.g. https://doi.org/10.1016/j.polymertesting.2019.106058.

Figure 5: It would be easier for reader if some arrows were added.

Line 195:’the fiber content” – there were layers of newspapers, not fibrils.

Line 207-211: This part needs to be rewritten and revised, so it makes more sense. What does it mean that “four layers of the test piece were bare”?

Line 238: “polypropylene plastic”- the same case as in comment for line 174

Line 433-436: I believe that this part can be confusing – such behavior was not observed in this research.

Line 443-445: Please remove this guideline.

Reviewer 2 Report

The article in my view is appropriate for a scientific journal such as Materials. In general the article is interesting and with a sufficient level of novelty. However, some aspects should be reviewed before publication. In my opinion is necessary a minor revision before its publication.

Introduction.

The authors describe briefly isotactic polypropylene, its applications and structure.

Secondly, they talk about compounds:

  • The authors should talk about reinforcement and not fillers. They refer to the incorporation of materials to improve mechanical properties and not to reduce the amount of plastic.
  • The sentence in line 40-41, is not correct. It is the strength of the composite material that is influenced by: Matrix properties, intrinsic properties of the reinforcement, dispersion of the reinforcement in the matrix, degree of orientation, quality of the interface and fraction in volume of reinforcement. Include references: (Sanadi et al., 1994; Thomason, 2002).

About the matrix: Cristallisation.

Fiber properties: aspect ratio, degree of crystallinity, etc...

Interface: fiber properties, type of bond generated, number of bonds. 

Line 46 add reference: Delgado-Aguilar et al., 2019.

When the authors talk about lignocellulosic fibers it is necessary to use the term fibers and not materials. (Line 66).

Materials and Methods.

Why have the authors decided to work with 3 %w/w MAPP? 

Results.

The chemical composition of the fibers, a key factor for the correct interface, should be included.

Lines 248-253. The analysis was performed on extracted fibers? Detail extraction procedure if applicable.

The experimentation carried out does not achieve the expected results, the authors should better justify the non-increase in mechanical properties.

Conclusions

The conclusions are presented and reflect the results obtained.

Delgado-Aguilar, M., Tarrés, Q., Marques, M. de F. V., Espinach, F.X., Julián, F., Mutjé, P., Vilaseca, F., 2019. Explorative Study on the Use of Curauá Reinforced Polypropylene Composites for the Automotive Industry. Materials (Basel). 12, 4185. doi:10.3390/ma12244185

Sanadi, A.R., Young, R.A., Clemons, C., Rowell, R.M., 1994. Recycled Newspaper Fibers as Reinforcing Fillers in Thermoplastics: Part I-Analysis of Tensile and Impact Properties in Polypropylene. J. Reinf. Plast. Compos. 13, 54–67. doi:10.1177/073168449401300104

Thomason, J.L., 2002. The influence of fibre length and concentration on the properties of glass fibre reinforced polypropylene: 5. Injection moulded long and short fibre PP. Compos. Part A Appl. Sci. Manuf. 33, 1641–1652. doi:10.1016/S1359-835X(02)00179-3

Reviewer 3 Report

It is rare that one receives for review an article so well written, such as that at hand. Reading it was a pleasure.
The article studies the effect of α and β nucleating agents on the crystallinity (i.e. crystallinity degree and type of crystal) and subsequently on the mechanical properties of polypropylene-fiber composites. The novel concept is that the nucleating agent here is "glued" on the surface of the fibers. The authors observe that the nucleating agents indeed promote trans crystallinity (considered beneficial for mechanical properties) and the β nucleating agent promotes the development of β crystallites (also considered beneficial), Nevertheless, the overall effect on mechanical properties is dominated by the nature of the fiber itself, whereas the effect of nucleating agent is rather minor.
The experiments are carefully described and the results are presented in an accurate way. The conclusions, are solidly supported by the results.
I recommend the article for publication, providing only some very minor comments:

- Experimental: In the results sections, results are reported for samples without fibres. So, in the experimental it would be good to mention those samples and the details of their preparation.

- section 2.2.1 Which adhesive was used?

- Table 1 It would be interesting to add a column with either the percent of β in total crsytallinity or the ratio β%/α%.

Reviewer 4 Report

This study entitled ‘Ligno-cellulosic fibre sized with nucleating agents promoting transcrystallinity in isotactic polypropylene composites’ is very confusing with its very poor English language. In addition, the objective and novelty of study are not clear and the manuscript is poorly presented. It is very difficult to follow the results since there is no flow throughout the manuscript. Furthermore, the discussion of manuscript needs significant improvement. Therefore, I must reject the manuscript for publication in Materials journal in its present form. The manuscript needs significant improvement (re-write) before it can be submitted elsewhere (any scientific journal) for publication.
